

# A longitudinal study of the impact of COVID-19 restrictions on students' health behavior, mental health and emotional well-being

Peter R. Reuter, Bridget L. Forster and Bethany J. Kruger

Marieb College of Health & Human Services, Florida Gulf Coast University, Fort Myers, FL, United States of America

## ABSTRACT

**Background**. COVID-related restrictions impacted the lives of students on and off campus during Academic Year 2020/2021.

**Methods**. Our study collected data on student health behavior and habits as well as their mental and emotional health using anonymous surveys. We compared these data with data collected prior to COVID in the longitudinal part of our study ($n = 721$) and analyzed them for the cross-sectional part of the study ($n = 506$).

**Results**. The longitudinal data show a significant difference for some student behaviors and habits, such as sleeping habits, physical activity, breakfast consumption, time spent online or playing video games, vaping, and marijuana use, during the COVID pandemic compared with pre-COVID data. Respondents also reported a significant increase in difficulty concentrating, remembering, or making decisions, as well as being impacted by feelings of sadness or hopelessness. Yet, there was no increase in the proportion of respondents considering, planning or attempting suicide during COVID. The cross-sectional data illuminate the negative effect of the overall situation and the restrictions on students' mental and emotional well-being. Three-quarters of respondents reported having craved human interaction during the past six months, more than half felt that their mental/emotional health had been impacted by the lack of social events or the switch to virtual (online) teaching. Two-thirds or more of respondents also expressed that they felt less connected to their peers and less motivated in their studies than in previous semesters. Fifty percent or more of respondents selected anxious, stressed, overwhelmed, disconnected, tired, and fatigued as words that best described their emotional state during the pandemic.

**Conclusions**. The impact of COVID-related restrictions on students' behaviors and habits as well as their mental and emotional health was less severe than one would have expected based on studies during the early stage of the pandemic. While some behaviors and habits changed during the COVID pandemic compared with the pre-COVID period, the changes were not substantial overall. Our study did not find an increase in the proportion of respondents considering, planning or attempting suicide during COVID, although the cross-sectional data from our survey make the negative effect of the overall situation and the restrictions on students' mental and emotional well-being evident. The impact of the pandemic will unquestionably be long-lasting and will necessitate further and future investigations.

Corresponding author
Peter R. Reuter, preuter@fgcu.edu

# INTRODUCTION

The COVID-19 pandemic started slowly and inconspicuously when the Wuhan Municipal Health Commission in China reported a cluster of pneumonia cases on December 31, 2019 (*World Health Organization, 2020*). Within one month, the case number had climbed to more than 9,800 worldwide and the death toll exceeded 200, prompting the World Health Organization (WHO) to declare a global health emergency. On March 11, 2020 the WHO declared COVID-19 a pandemic (*World Health Organization, 2021*). Within a few days, many countries went into some form of lockdown, and most public and private colleges and universities were forced to either shut down temporarily or to transition to online (remote) learning. These sudden and unexpected changes impacted millions of college and university students around the world. Many were forced to move back home or find other alternative living arrangements when campuses and dorms closed. The shift to online teaching also created challenges and struggles for many students as it required a level of hardware, software, and internet connectivity for which many students were not prepared (*Aucejo et al., 2020*; *Gillis & Krull, 2020*).

Although the situation differed for students across different countries, studies looking at the impact in the early part of the pandemic (Spring and Summer 2020) equally reported that students experienced high levels of stress that negatively impacted their mental health leading to increased levels of anxiety and depression (*Essadek & Rabeyron, 2020*; *Kaparounaki et al., 2020*; *Saddik et al., 2020*; *Wathelet et al., 2020*; *Zandifar & Badrfam, 2020*; *Zhang et al., 2020*; *Baloch et al., 2021*; *Ochnik et al., 2021*). Longitudinal studies that compared data collected from before the pandemic (pre-COVID) with data collected during the pandemic reported similar results. For example, data collected at a university in the United Kingdom in Fall 2019 and Spring 2020 by *Evans et al. (2021)* showed that students experienced a decrease in well-being and increase in depression. *Savage et al. (2020)* also collected data from students at a university in the United Kingdom in Fall 2019 and Spring 2020; they too found increased stress levels and a decrease in mental well-being of participating students.

Studies involving college and university students in the US also reported increases in anxiety and depression during this period (*Holingue et al., 2020*; *Jeffery & Bauer, 2020*; *Kecojevic et al., 2020*; *Wang et al., 2020*; *Hawes et al., 2021*). Again, the results of cross-sectional studies were confirmed by longitudinal studies such as the studies by *Fruehwirth, Biswas & Perreira (2021)*, *Giuntella et al. (2021)*, and *Wilson et al. (2021)*. The study with the largest number of participants in the US was conducted by the Healthy Minds Network (HMN) in association with the American College Health Association (ACHA); more than 18,000 students on 14 campuses participated from late March through May 2020 (*Martinez & Nguyen, 2020*). Two-thirds of participants reported experiencing more stress, and one-third had gone through a change in living arrangements. The participants also

reported an increase in depression, and 60% indicated that the pandemic had made it more difficult to access mental health care providers.

There are, however, only a few studies exploring how the sudden changes and restrictions impacted students' health behaviors and habits. *Bertrand et al. (2021)* collected data from students at two universities in Canada in April/May 2020. They found increased sedentary behavior and alcohol consumption, a reduction in physical activity, and a drop in the quality of the students' diet. A study from the UK comparing pre-COVID (Fall 2019) and COVID (April/May 2020) data reported a decrease in alcohol consumption and a shift in sleep behavior; participants reported staying up later but not sleeping longer (*Savage et al., 2020*). In the US, *Moriarty et al. (2021)* found that while some students exercised more and slept longer, other students did just the opposite, while *Maher et al. (2021)* reported a loss of sleep quality.

Studies from different countries described a decline in physical activity of students during this first part of the pandemic (*Savage et al., 2020*; *Fruehwirth, Biswas & Perreira, 2021*; *Giuntella et al., 2021*; *López-Valenciano et al., 2021*; *Maher et al., 2021*). However, two studies from Spain reported opposite results; in the study by *Romero-Blanco et al. (2020)*, physical activity increased during the early lockdown, whereas *Castañeda Babarro et al. (2020)* found a decrease in physical activity during the same period. *Du et al. (2021)* conducted a multinational study involving students in Asia, Europe, and North America that investigated the impact of the pandemic on sleep quality, dietary behavior, and alcohol consumption. They showed a negative impact on the diet of participants and sleep quality; nonetheless, there was no increase in alcohol consumption. Participants in *Firkey, Sheinfil & Woolf-King*'s (*2021*) study reported an increase in anxiety and a decrease in quality of life and sleep quality; nonetheless, there was no increase in alcohol or marijuana consumption associated with the change in mental health. *Sokolovsky et al. (2021)* reported a decrease in the frequency of smoking and vaping among students in early Spring 2020.

Once the early stage of the pandemic had passed, different countries adopted diverging approaches depending on the number of COVID cases as well as economic and political considerations (*Ibrahim, Binofai & Alshamsi, 2020*; *OECD, 2020*). Some countries went through extended lockdowns while others decided against further restrictions or used shorter complete or partial lockdowns only. The United States was a mirror image of what happened around the world; some states imposed extended lockdowns that included closure of schools and universities, others decided to open up everything as soon as possible (*Treisman, 2020*). The differences in when and to what extent countries were affected by the second and third wave of the pandemic and which approach they adopted are the main reasons why there are only a few longitudinal studies on the impact on students' mental health and health behavior beyond the initial stage so far. *Savage et al. (2021)* investigated the impact on mental health and physical activity of university students in the UK nine months into the pandemic. Study participants still reported reduced mental well-being and physical activity, and increased perceived stress and sedentary behavior in October 2020. Collecting data from graduating students at a university in Ethiopia in November 2020, *Mekonen et al. (2021)* found that one-fifth of students reported stress, and two-fifths anxiety and depression.

Our research project on student health behaviors and habits started in Spring 2018 and, therefore, adding survey data from Fall 2020 and Spring 2021 enabled us to assess the impact COVID-related restrictions had on our students beyond the initial phase. Although the situation in Florida and at our university was different from what students in other countries and at other institutions of higher education experienced, the results are an important contribution to our understanding of how prolonged extreme events can impact students' behaviors, habits, and mental and emotional health.

## METHODS

Data for this article were collected between April 2018 and April 2021 using two anonymous online surveys. Each Summer and Fall semester, emails were sent to students enrolled in courses that are taken by students from different colleges and majors, such as 'Statistical methods' or 'Introduction into Psychology', inviting them to participate in our survey. The total number of students invited per semester was approximately 2,000–2,500. The number of students participating in the survey per semester usually ranged from 150–250, *i.e.,* the response rate was in the range of 6–10%.

The first survey, used from April 2018 to February 2020, consisted of five groups of questions around health and wellness (see Appendix 1); the survey was described previously (*Reuter & Forster, 2021*). Some of the data collected, but unrelated to the current research, has been published elsewhere (*Reuter, Forster & Brister, 2020*; *Reuter & Forster, 2021*).

After the onset of the COVID-19 pandemic and the introduction of pandemic-related restrictions nationally, locally, and at Florida Gulf Coast University (FGCU), the survey was modified in order to capture the impact these restrictions had on students' health behaviors, and habits as well as their mental and emotional health (see Appendix 2).

Although the study used a repeated cross-sectional survey design, the data collected did not consists of independent data sets as students may have participated more than once during different semesters. Thus, our study contained repeated cross-sectional and panel elements.

Both surveys were approved by the Institutional Review Board (IRB) at FGCU prior to sending email invitations to students (IRB 2018-17). The first page of the survey consisted of an approved online survey consent form; in other words, consent was obtained. Participation was voluntary and participants did not receive any compensation or any other direct or indirect benefit from the university or us.

### COVID-19 related restrictions in Florida and at FGCU

Florida went into a 30 day stay-at-home order in April 2020, with Phase 1 reopening already taking effect by mid-May for all counties (*DeSantis, 2020*). Phase 2, which included restaurants operating at 50%, gyms at full capacity with social distancing in place, gatherings expanded to a 50 person limit, and employers beginning to develop a plan to bring workers back, was put in place by mid-June 2020. Phase 3, which began in September 2020, removed many of the restrictions left from phase 2, and also allowed non-essential travel, full capacity at restaurants and bars, and gyms and recreation centers operating at full capacity without social distancing enforced. The State University System of Florida Blueprint for Reopening

Campuses for Fall Semester 2020 focused on promoting a healthy campus and community environment while still offering successful academic program delivery (*University System of Florida, 2020*).

FGCU is a regional university located in Southwest Florida with a student population of about 15,000 undergraduate and graduate students (*Florida Gulf Coast University, 2021a*). Ninety-seven percent of students are from Florida and half of them come from Southwest Florida. Six out of ten students identify as White, slightly more than one-fifth as Hispanic, and one in twelve as Black. Students of Asian origin make up 3.3% of the student body, and Native American students account for 0.9% (*Florida Gulf Coast University, 2021a*). FGCU's COVID plan for the Fall semester 2020 was finalized in early June (*Florida Gulf Coast University, 2020a*). Many of the new rules impacted students' lives, including limited shuttle routes and seating on buses, mask requirements everywhere on campus, and enforced social distancing. Additional measures included restrictions and limitations on intramural sports, closure of indoor and outdoor recreational facilities, and group events and meetings capped at 50%. All housing units were adjusted for single bedroom occupancy, and class and classroom sizes were reduced (about 38% of campus space utilized at one time with 50% of classroom capacity). Slightly more than 50% of classes offered were online classes (28% online asynchronous, 24% online synchronous), approximately 45% of classes were taught in the classroom, and less than 4% were scheduled as hybrid classes. The COVID-19 plan for Spring and Summer 2021 was an extension of the plan for Fall 2020 (*Florida Gulf Coast University, 2020b*).

## Data analysis

Due to the voluntary nature of the survey, sample sizes vary for different analyses, but are included in the results description or tables. Data are presented as means with standard deviation for questions with quantitative answers and as a percentage of the total participant pool, or a portion of this pool, for questions with categorical answer options.

We used Pearson Chi-Square Tests, Fisher's Exact Test, and Kruskall-Wallis Rank Sums Tests to examine differences between the pre-COVID and COVID survey group. The dependent variable in all tests was survey group (a categorical variable with two options: pre-COVID and COVID). Independent variables are listed in Table 1 below.

We used the Benjamini–Hochberg Procedure (with a false discovery rate of 5%) to determine significance due to the number of statistical analyses performed. Using this approach, tests with a *p*-value <0.0171 are considered significant. Statistical analyses were performed using JMP software program Version 16 (JMP®; SAS Institute Inc., Cary, NC, USA).

# RESULTS

## Demographic characteristics of respondents

After excluding responses that indicated an age of less than 18 years or did not provide an age, we were left with 721 responses for the pre-COVID survey and 506 responses for the COVID survey (Table 2). There was no difference between the two survey groups in the proportion of freshmen, sophomores, juniors, seniors, or other students (graduate,

**Table 1  Independent variables included in the study and data analysis.**

| Behaviors and habits | Independent variable |
| --- | --- |
| **Sleeping habits** | |
| Hours of sleep per night | Average number of hours slept per night (categorical data: 4 h or less; 5 h; 6 h; 7 h; 8 h; 9 h; 10 h or more per night) |
| Average time going to bed | Average time of night going to bed (categorical data: before 8pm; between 8pm and 10pm; between 10pm and 12am; after 2am) |
| **Working** | |
| Do you work? | Categorical data (Yes; No) |
| Average hours worked per week | Average number of hours worked per week (categorical data: [including all students who did not work]; under 5 h; 5–10 h; 10–20 h; 20–30 h; 30–40 h; over 40 h) |
| **Physical activity** | |
| Physical activity | Number of days per week with physical activity of at least 60 min (continuous numerical data, range: 0 to 7 days) |
| Strength training | Number of days per week doing exercises to strengthen or tone muscles (continuous numerical data, range: 0 to 7 days) |
| **Eating habits** | |
| Consumption of vegetables | Number of times vegetables consumed in past 7 days (categorical data: 0 times; 1 to 3 times; 4 to 6 times; 7 to 10 times; 11 times or more) |
| Consumption of fruit | Number of times fruit consumed in past 7 days (categorical data: 0 times; 1 to 3 times; 4 to 6 times; 7 to 10 times; 11 times or more) |
| Breakfast consumption | Number of days breakfast eaten in past 7 days (continuous numerical data, range: 0 to 7 days) |
| Fast food consumption | Number of times fast food consumed in past 7 days (categorical data: 0 times; 1 to 3 times; 4 to 6 times; 7 to 10 times; 11 times or more) |
| Coffee consumption | Number of cups of coffee consumed per day (categorical data: 0 cups, 1-2 cups, 3-4 cups, 5-6 cups, 7 cups or more) |
| **Online activity** | |
| Time spent watching TV, movies, reading news online | Number of hours per day (categorical data: 0 h, less than 1 h, 1 h, 2 h, 3 h, 4 h, 5 h or more) |
| Time spent playing video or computer games | Number of hours per day (categorical data: 0 h, less than 1 h, 1 h, 2 h, 3 h, 4 h, 5 h or more) |
| **Vaping, alcohol and marijuana** | |
| Vaping | Number of days vaping in past 7 days (continuous numerical data, range: 0 to 7 days) |
| Alcohol consumption | Number of days consuming at least one drink during the last 30 days (categorical data: 0 days, 1–2 days, 3–5 days, 6–9 days, 10–19 days, 20–29 days, all 30 days) |
| Marijuana use | Number of days using marijuana in past 7 days (continuous numerical data, range: 0 to 7 days) |

**Table 1** (*continued*)

| Behaviors and habits | Independent variable |
| --- | --- |
| **Mental/emotional health** | |
| Difficulties concentrating[a] | Categorical data (Yes; No) |
| Feeling sad or hopeless[b] | Categorical data (Yes; No) |
| Considered suicide[c] | Categorical data (Yes; No) |
| Number of times suicide considered[d] | Number of times over past 12 months (categorical data: 1 time, 2-3 times, 4-5 times, 6 times or more) |
| Planned suicide[e] | Categorical data (Yes; No) |
| Attempted suicide[f] | Categorical data (Yes; No) |

Notes.
[a] Do you have serious difficulty concentrating, remembering, or making decisions because of a physical, mental, or emotional problem?
[b] During the past 12 months, did you ever feel so sad or hopeless almost every day for two weeks or more in a row that you stopped doing some usual activities?
[c] During the past 12 months, did you ever seriously consider attempting suicide?
[d] How many times did you consider attempting suicide?
[e] Did you make a plan about how you would attempt suicide?
[f] Did you actually attempt suicide?

non-degree seeking, and second-degree seeking students) (Pearson Chi-Square Test, Chi-square = 2.637, DF = 5, $p = 0.76$). There was, however, a change in living circumstances between the two survey groups (Pearson Chi-Square Test, Chi-square = 22.239, DF = 3, $p < 0.0001$), with more students living at home and fewer students living on-campus in the COVID group. In both survey groups, female respondents outnumbered male respondents, and the COVID group had proportionally more female respondents than the pre-COVID group (Fisher's Exact Test, $p = 0.0088$), even though the student body at the university consists of 52% female and 48% male students (*Florida Gulf Coast University, 2021b*).

## Sleeping habits

The proportion of respondents reporting average hours of sleep over the past seven days differed between the pre-COVID group and the COVID group (Pearson Chi-Square Test, Chi-square = 29.890, DF = 6, $p < 0.0001$). The percentage of respondents reporting to sleep eight hours or more was almost 50% higher for the COVID group (30.6%) compared with the pre-COVID group (21.1%).

The time when respondents went to bed also changed between the pre-COVID group and the COVID group, as the proportion of respondents reporting an average bedtime differed between the pre-COVID and the COVID group (Pearson Chi-Square Test, Chi-square = 22.942, DF = 4, $p < 0.0001$). In the COVID group, more respondents reported going to sleep after 2 am and fewer students went to bed between 10 pm and 2 am.

## Working

There was no difference in the proportion of respondents that indicated they worked zero hours per week between the pre-COVID group (45.3%) and the COVID group. (50.3%; Pearson Chi-Square Test, Chi-square = 3.452, DF = 1, $p = 0.0632$). Among respondents that worked at least one hour per week on average ($n = 395$ in the pre-COVID group; $n = 251$ in the COVID group), there was also no difference in the proportion of respondents reporting average hours worked.

**Table 2** Demographic characteristics of respondents for the pre-COVID (data collected between 04/2018 and 02/2020) and the COVID group (data collected between 11/2020 and 04/2021).

|  | pre-COVID(04/2018–02/2020) | COVID(11/2020–04/2021) |
|---|---|---|
| **Number of respondents** | 721 | 506 |
| **Biological sex** |  |  |
| Female | 77.4% | 83.8% |
| Male | 21.4% | 16.0% |
| No information | 1.3% | 0.2% |
| **Age** (mean ± std. dev; range; median age) | 20.3 ± 4.1; 18–61; 19 | 20.6 ± 4.8; 18–65; 19 |
| **Student population** |  |  |
| Freshman | 32.2% | 31.4% |
| Sophomore | 27.2% | 31.0% |
| Junior | 24.4% | 22.3% |
| Senior | 13.0% | 12.7% |
| Other (non-degree/second-degree seeking, graduate students) | 2.4% | 1.8% |
| No information | 0.8% | 0.8% |
| **Living arrangements** |  |  |
| At home | 22.5% | 32.2% |
| On campus | 51.2% | 38.5% |
| Off campus with others | 17.2% | 20.0% |
| Off campus alone | 9.2% | 9.3% |

## Physical activity

There were differences in the self-reported physical activity between the two survey groups (Table 3). Respondents in the COVID group reported a lower average number of days of physical activity per week (2.7 ± 2.2, mean ± std. dev) than respondents in the pre-COVID group (3.1 ± 2.2, mean ± std. dev) (Kruskal–Wallis Rank Sums Test, Chi-square = 9.3421, DF = 1, $p = 0.0022$). Respondents from the COVID group reported less physical activity with more students not being physically active at all (21.4% *vs.* 16.6% pre-COVID) and fewer students being physically active on six or seven days (11.6% *vs.* 18.2% pre-COVID). Respondents in the COVID group also reported a lower average number of days of strength training per week (1.7 ± 2.1, mean ± std. dev) than respondents in the pre-COVID group (1.9 ± 2.1, mean ± std. dev) (Kruskal–Wallis Rank Sums Test, Chi-square = 2.2429, DF = 1, $p = 0.1342$.

## Eating habits

There were no differences in the consumption of vegetables and fruits between the two survey groups (Table 3). The proportion of respondents reporting frequency of vegetable consumption over a 7-day period did not differ between the pre-COVID group and the COVID group (Pearson Chi-Square Test, Chi-square = 1.591, DF = 4, $p = 0.8104$). The proportion of respondents reporting frequency of fruit consumption over a 7-day period also did not differ between the pre-COVID group and the COVID group (Pearson Chi-Square Test, Chi-square = 2.726, DF = 4, $p = 0.6046$).

**Table 3** Independent variables, number of respondents for each survey group, statistical test used, and adjusted p-values using the Benjamini-Hochberg procedure.

| Independent variable | Number of respondents | | Test | p-value |
|---|---|---|---|---|
| | pre-COVID | COVID | | |
| Hours of sleep per night | 721 | 506 | Pearson Chi-Square Test, Chi-square = 29.890, DF = 6 | **0.0001**[*] |
| Average time going to bed | 721 | 506 | Pearson Chi-Square Test, Chi-square = 22.942, DF = 4 | **0.0001**[*] |
| Do you work? | 721 | 506 | Pearson Chi-Square Test, Chi-square = 3.452, DF = 1 | 0.0632 |
| Average hours worked per week | 395 | 251 | Pearson Chi-Square Test, Chi-square = 5.510, DF = 5 | 0.3568 |
| Physical activity (aerobic exercise) | 693 | 500 | Kruskal–Wallis Rank Sums Test, Chi-square = 9.3421, DF = 1 | **0.0022**[*] |
| Strength training | 693 | 500 | Kruskal–Wallis Rank Sums Test, Chi-square = 2.2429, DF = 1 | 0.1342 |
| Consumption of vegetables | 693 | 498 | Pearson Chi-Square Test, Chi-square = 1.591, DF = 4 | 0.8104 |
| Consumption of fruit | 692 | 498 | Pearson Chi-Square Test, Chi-square = 2.726, DF = 4 | 0.6046 |
| Breakfast consumption | 692 | 499 | Kruskal–Wallis Rank Sums Test, Chi-square = 7.7713, DF = 1 | **0.0053**[*] |
| Eating fast food Yes/No | 693 | 497 | Pearson Chi-Square Test, Chi-square = 4.513, DF = 1 | 0.0336 |
| Fast food consumption | 693 | 497 | Pearson Chi-Square Test, Chi-square = 4.417, DF = 3 | 0.2198 |
| Drinking coffee Yes/no | 690 | 500 | Pearson Chi-Square Test, Chi-square = 3.219, DF = 1 | 0.0728 |
| Coffee consumption | 690 | 500 | Pearson Chi-Square Test, Chi-square = 5.229, DF = 3 | 0.1558 |
| Time spent watching TV, movies, reading news online | 721 | 505 | Pearson Chi-Square Test, Chi-square = 23.071, DF = 6 | **0.0008**[*] |
| Time spent playing video or computer games | 718 | 506 | Pearson Chi-Square Test, Chi-square = 36.963, DF = 6 | **0.0001**[*] |
| Vaping | 236 | 185 | Kruskal–Wallis Rank Sums Test, Chi-square = 11.3200, DF = 1 | **0.0008**[*] |
| Drinking alcohol Yes/No | 523 | 499 | Pearson Chi-Square Test, Chi-square = 65.505, DF = 1 | **0.0001**[*] |
| Alcohol consumption | 523 | 499 | Pearson Chi-Square Test, Chi-square = 6.649, DF = 4 | 0.1556 |
| Marijuana use | 292 | 500 | Kruskal–Wallis Rank Sums Test, Chi-square = 15.6320, DF = 1 | **0.0001**[*] |
| Difficulties concentrating | 633 | 502 | Pearson Chi-Square Test, Chi-square = 77.283, DF = 1 | **0.0001**[*] |
| Feeling sad or hopeless | 625 | 502 | Pearson Chi-Square Test, Chi-square = 5.686, DF = 1 | **0.0171**[*] |
| Considered suicide | 619 | 491 | Pearson Chi-Square Test, Chi-square = 0.259, DF = 1 | 0.6108 |

**Table 3** (*continued*)

| Independent variable | Number of respondents | | Test | *p*-value |
|---|---|---|---|---|
| | pre-COVID | COVID | | |
| Number of times suicide considered | 78 | 59 | Pearson Chi-Square Test, Chi-square = 1.399, DF = 3 | 0.7057 |
| Planned suicide | 78 | 59 | Pearson Chi-Square Test, Chi-square = 2.762, DF = 3 | 0.0965 |
| Attempted suicide | 34 | 36 | Pearson Chi-Square Test, Chi-square = 0.229, DF = 1 | 0.6324 |

**Notes.**
*Denotes statistical significance.

However, respondents in the COVID group reported a lower average number of days per week they ate breakfast ($3.7 \pm 2.7$, mean $\pm$ std. dev) compared with respondents in the pre-COVID group ($4.1 \pm 2.5$, mean $\pm$ std. dev) (Kruskal–Wallis Rank Sums Test, Chi-square = 7.7713, DF = 1, $p = 0.0053$). Students in the COVID group were twice as likely to skip breakfast as students in the pre-COVID group (19.0% *vs.* 9.8%).

The proportion of respondents who reported eating fast food at least once in the past seven days did not differ in the COVID group (78.3%) from the pre-COVID group (72.3%) ((Pearson Chi-Square Test, Chi-square = 4.513, DF = 1, $p = 0.0336$). Additionally, for those that reported eating fast food, the amount of times fast food was consumed over a 7-day period did not differ between the pre-COVID group and the COVID group (Pearson Chi-Square Test, Chi-square = 4.417, DF = 3, $p = 0.2198$).

There were no differences in the consumption of cups of caffeinated coffee between the pre-COVID and the COVID group (Pearson Chi-Square Test, Chi-square = 3.219, DF = 1, $p = 0.0728$). The proportion of respondents who reported drinking caffeinated coffee at least once in the past seven days did not differ between the two groups either. Among those who drank coffee at least once, the proportion of respondents reporting frequency of caffeinated coffee consumption over a 7-day period did not differ between the two groups Pearson Chi-Square Test, Chi-square = 5.229, DF = 3, $p = 0.1558$).

## Online activity
The proportion of respondents reporting the amount of time spent watching television or reading online over a 7-day period differed between the pre-COVID and the COVID group (Pearson Chi-Square Test, Chi-square = 23.071, DF = 6, $p = 0.0008$). Almost one-quarter of respondents from the COVID group (23.8%) spent five hours or more online watching TV or movies per day compared with one in seven respondents of the pre-COVID group (14.6%).

Likewise, there was a difference between the two survey groups in the proportion of respondents reporting the amount of time spent playing video games or using an electronic device for things that are not school work over a 7-day period (Pearson Chi-Square Test, Chi-square = 36.963, DF = 6, $p < 0.0001$).

## Vaping, alcohol, and marijuana
Respondents in the COVID group reported vaping on a higher average number of days per week ($2.9 \pm 3.2$ days per week, mean $\pm$ std. dev) than respondents in the pre-COVID

group (1.6 ± 2.5 days per week) (Kruskal–Wallis Rank Sums Test, Chi-square = 11.3200, DF = 1, $p = 0.0008$). Nonetheless, vaping did not become more prevalent overall during the COVID pandemic; 35.5% of survey respondents from the pre-COVID group and 37.1% of respondents from the COVID group had vaped before. Conversely, one in three respondents from the COVID group (34.6%) reported having vaped daily compared with one in seven respondents from the pre-COVID group (15.7%).

The proportion of respondents who reported drinking alcohol on at least one day in the last 30 days was lower in the COVID group (50.5%) compared with the pre-COVID group (75.0%) (Pearson Chi-Square Test, Chi-square = 65.505, DF = 1, $p < 0.0001$). However, of those that did drink at least one day in the prior month, the number of days on which alcohol was consumed did not differ between the pre-COVID group and the COVID group.

Respondents in the COVID group reported using marijuana on a higher average number of days per week (1.5 ± 2.4 days per week, mean ± std. dev) than respondents in the pre-COVID group (1.0 ± 2.3 days per week; Kruskal–Wallis Rank Sums Test, Chi-square = 15.6320, DF = 1, $p < 0.0001$). Additionally, more respondents from the COVID group than the pre-COVID group had used marijuana during the past seven days (20.2% *vs.* 15.0%) and more than twice as many of them had used every day during that time (10.0% *vs.* 4.2%).

## Mental/emotional health

A higher proportion of respondents from the COVID group reported having serious difficulty concentrating, remembering, or making decisions compared with the pre-COVID group (47.2% *vs.* 22.4%; Pearson Chi-Square Test, Chi-square = 77.283, DF = 1, $p < 0.0001$). There was also an increase in the proportion of respondents in the COVID group who had felt so sad or hopeless almost every day for two weeks or more in a row that they stopped doing some usual activities (44.6% in the COVID group *vs.* 37.6% in the pre-COVID group; Pearson Chi-Square Test, Chi-square = 5.686, DF = 1, $p = 0.0171$).

Conversely, there was no change in the proportion of respondents who had seriously considered committing suicide (12.2% in the COVID group *vs.* 13.3% in the pre-COVID group; Pearson Chi-Square Test, Chi-square = 0.259, DF = 1, $p = 0.6108$). Among those who had considered committing suicide, there was no change between the survey groups in the number of times they had considered attempting suicide (Pearson Chi-Square Test, Chi-square = 1.399, DF = 3, $p = 0.7057$). In regards to acting on thoughts about suicide, a similar proportion of students in both survey groups planned out their suicide attempt (47% in the pre-COVID group; 62% in the COVID group; Pearson Chi-Square Test, Chi-square = 2.762, DF = 3, $p = 0.0965$). Of these, a similar proportion (18% in the pre-COVID group and 22% in the COVID group) actually did attempt suicide (Pearson Chi-Square Test, Chi-square = 0.229, DF = 1, $p = 0.6324$).

The survey used during the COVID pandemic from November 2020 to April 2021 contained additional questions to gather information on students' emotional state (Table 4). One-quarter of respondents indicated not feeling safe on campus during COVID, and three-quarters of participating students said that they had craved human interaction over

**Table 4  Questions exploring how COVID-related changes affected the emotional health of students with answer options.**

| Questions and answer options | Responses |
|---|---|
| Do you feel safe on campus this semester? | $n = 514$ |
|     Yes | 24.7% |
|     No | 75.3% |
| During the past 6 months have you felt yourself craving human interaction? | $n = 519$ |
|     Yes | 76.7% |
|     No | 23.3% |
| Do you think the lack of social events this semester has affected your mental/emotional health? | $n = 518$ |
|     Yes | 54.4% |
|     No | 45.6% |
| Has going to mostly virtual teaching affected your mental/emotional health? | $n = 517$ |
|     Yes | 55.7% |
|     No | 44.3% |
| In comparison to previous semesters, do you find yourself more or less connected to your peers?[*] | $n = 355$ |
|     More | 6.2% |
|     About the same | 21.4% |
|     Less | 72.4% |
| How motivated do you feel in your classes this semester compared with previous semesters?[*] | $n = 355$ |
|     More | 7.6% |
|     About the same | 25.6% |
|     Less | 66.8% |

**Notes.**
[*]Only students enrolled at FGCU during the previous semester were asked to respond.

the previous six months. More than half of respondents felt that their mental/emotional health had been impacted by the lack of social events or the switch to virtual (online) teaching. Two-thirds or more of respondents also expressed that they felt less connected to their peers and less motivated in their studies than in previous semesters.

Furthermore, participants were also invited to select up to ten words that described their emotional state from a list of thirty-nine words (Table 5). Overall 504 students participated, although not all respondents selected a full set of ten words. The six emotions selected by more than 50% of respondents were anxious, stressed, overwhelmed, disconnected, tired, and fatigued. At the other end of the rankings, the six emotions selected by the smallest percentage of respondents were sanguine, serene, upbeat, inspired, peaceful, and frightened.

# DISCUSSION

The purpose of this research project was to study the impact prolonged restrictions related to COVID had on students' health behavior as well as their mental and emotional

**Table 5 Ranking of emotions by percentage of respondents selecting up to ten words from a list of 39 words describing emotions ($n = 504$).**

| Rank | Emotion | % | Rank | Emotion | % | Rank | Emotion | % |
|---|---|---|---|---|---|---|---|---|
| 1 | Anxious | 70.0 | 14 | Gloomy | 21.4 | 27 | Confident | 14.0 |
| 2 | Stressed | 64.6 | 15 | Optimistic | 20.6 | 28 | Miserable | 13.8 |
| 3 | Overwhelmed | 58.8 | 16 | Neutral | 19.3 | 29 | Hopeless | 13.6 |
| 4 | Disconnected | 58.4 | 17 | Sad | 18.3 | 30 | Engaged | 10.9 |
| 5 | Tired | 54.9 | 18 | Indifferent | 18.1 | 31 | Demoralized | 9.9 |
| 6 | Fatigued | 53.1 | 19 | Negative | 17.9 | 32 | Energized | 9.0 |
| 7 | Alone | 47.9 | 20 | Unhappy | 16.5 | 33 | Relaxed | 8.6 |
| 8 | Depressed | 38.9 | 21 | Distressed | 16.3 | 34 | Frightened | 8.2 |
| 9 | Withdrawn | 33.9 | 22 | Angry | 16.1 | 35 | Peaceful | 8.2 |
| 10 | Hopeful | 32.3 | 23 | Happy | 15.8 | 36 | Inspired | 7.0 |
| 11 | Lonely | 30.4 | 24 | Positive | 15.8 | 37 | Upbeat | 3.5 |
| 12 | Moody | 30.4 | 25 | Calm | 15.4 | 38 | Serene | 1.4 |
| 13 | Empty | 25.9 | 26 | Powerless | 14.6 | 39 | Sanguine | 1.2 |

health. Because the situation was unique in recent history, the results add to our general understanding of how extreme events impact people's physical and mental health. The longitudinal data of our study show a significant difference for some student behaviors and habits, such as sleeping habits, physical activity, breakfast consumption, time spent online or playing video games, vaping, and marijuana use, during the COVID pandemic compared with pre-COVID data. While study respondents also reported a significant increase in difficulty concentrating, remembering, or making decisions, as well as being impacted by feeling sad or hopeless, there was no increase in the proportion of respondents considering, planning or attempting suicide during COVID.

The cross-sectional data from the survey administered during the COVID pandemic illuminate the negative effect of the overall situation and the restrictions on students' mental and emotional well-being. While we have no pre-COVID data to compare with, it is doubtful that three-quarters of respondents would have craved human interaction during normal times. It is also telling that the six words describing emotions selected by half of respondents or more describe negative emotional states. On the other hand, respondents in our survey were not more worried about their safety on campus during COVID. Data from the *American College Health Association (2020)* show that prior to COVID one in seven male students and one in four female students did not feel safe on campus during daytime hours, and one in two male and four in five female students did not feel safe on campus at night.

The change in the proportion of students living at home can be attributed to a number of factors. Although more than half of respondents (51.3%) were enrolled in at least one in-person face-2-face class, most of them took mainly online classes. Living at home and commuting once or twice per week to campus was preferable at a time where many families experienced economic hardship. Plus, students living at home faced fewer restrictions on

social interactions and outside activities than students living on campus, making living at home more attractive.

The fact that there is no difference in the proportion of students working and in the number of hours worked is a reflection of the overall approach to handling the pandemic by the state of Florida. The difference in sleeping patterns found in our study with students sleeping longer and staying up later during the COVID pandemic do not point at a loss of sleep quality. With fewer students having to get up in the morning to go to campus or to get to class, one can understand that there is a tendency to stay up late and sleep longer. Thus, our findings do not confirm the loss of sleep quality reported by *Deng et al. (2021)*, *Du et al. (2021)*, *Firkey, Sheinfil & Woolf-King (2021)*, *Giuntella et al. (2021)*, and *Maher et al. (2021)* during the early (lockdown) stage in different countries. The study by *Moriarty et al. (2021)* was conducted in the US in June/July 2020 and its results are similar to ours. *Evans et al. (2021)* also found a shift to staying up late in their study of undergraduate psychology students at a UK university.

The same reasons, *i.e.,* not having to get up in the morning, as well as the limitations on social activities, may explain shifts in sleeping patterns and why respondents reported more screen time during the COVID pandemic (*i.e.,* spending more time online reading or watching TV or movies, and playing video or computer games). *Giuntella et al. (2021)* also reported an increase in screen time. This increase can also be considered increased sedentary behavior and confirms similar findings reported by *Savage et al. (2020)* and *Bertrand et al. (2021)* in the early pandemic stage as well as in the longitudinal study by *Savage et al. (2021)*.

The decline in physical activity reported by our study respondents from pre-COVID to COVID is in line with results for the early pandemic published by *Savage et al. (2020)*, *Bertrand et al. (2021)*, *Giuntella et al. (2021)*, *López-Valenciano et al., (2021)*, and *Wilson et al. (2021)*. The longitudinal study by *Savage et al. (2021)* also reported a decrease in physical activity among university students in the UK nine months into the pandemic.

While we found a negative impact on breakfast and fast-food consumption, this should not be understood as a reduction in the quality of the diet along the line of results reported by *Bertrand et al. (2021)* and *Du et al. (2021)*. These changes may be due to students' changed living arrangements and sleeping habits. For example, they may skip breakfast because they sleep longer or may consider the first meal of the day to be lunch if they sleep in. Likewise, because more of them live at home without a meal plan, they may be prone to picking up fast food on the way to or from work and campus.

The increase in vaping, alcohol consumption, and marijuana use from the pre-COVID to the COVID period aligns with the increases found in studies during the early pandemic (*Bertrand et al., 2021*; *Du et al., 2021*; *Evans et al., 2021*; *Firkey, Sheinfil & Woolf-King, 2021*). Contrary to these studies and ours, the retrospective study by *Sokolovsky et al. (2021)* found that the frequency of smoking and vaping decreased after students went into lockdown.

The longitudinal as well as the cross-sectional survey data in our study show an increase in stress levels, anxiety, and depression during the COVID period, and thus, broadly confirm studies from the early pandemic involving college and university students in

the US (*Holingue et al., 2020*; *Jeffery & Bauer, 2020*; *Kecojevic et al., 2020*; *Wang et al., 2020*; *Hawes et al., 2021*) and other countries (*Essadek & Rabeyron, 2020*; *Kaparounaki et al., 2020*; *Wathelet et al., 2020*; *Baloch et al., 2021*). Fortunately, our data show that the proportion of students considering, planning, and attempting suicide did not increase during the pandemic. This was also reported by *Pirkis et al. (2021)*, who analyzed data from 21 countries around the world in Fall 2020 and found no evidence for an increase in suicide rates compared with pre-pandemic levels.

A major strength of our study is having data from before the COVID pandemic through the later part of the pandemic. Even though the data are not truly longitudinal, this enabled us to compare how students' health behaviors and habits changed or did not change during the course of the pandemic, and how students coped with the extended period of stress and uncertainty. The main limitations of our study were participant selection and reliance on self-reported health behaviors and habits. Even though students from all colleges across the university were invited to participate, the overall response rate was in the range of 6–10% only. We received almost 80% of responses from female students, whereas the FGCU student body has only slightly more than 50% female students (*Florida Gulf Coast University, 2021b*). Additionally, we cannot rule out that students may have participated more than once during the same semester.

Every survey has to question whether respondents may have intentionally or unintentionally provided incorrect information, especially when behaviors are involved that are not legal or socially acceptable. For example, the legal drinking age in Florida is 21 years and, therefore, students under 21 who report drinking alcohol admit to breaking the law. The same applies to the use of recreational drugs. Also, some of the health behaviors and habits included in the survey were not well defined and respondents may have interpreted them differently. For instance, respondents may have considered walking to class as being physically active. We are also aware that some questions could be considered overlapping; for example, the questions relating to time spent watching TV, movies, or reading news online, and time spent playing video or computer games. We also failed to include a question relating to time spent on social media in general in our survey, leading to responses indicating that students had not spent much time online overall. However, because we looked at a change from pre-COVID to COVID, this mistake did not impact our results negatively.

One also has to point out that Florida had very limited restrictions compared with other states and countries, which led to our respondents being able to continue to work almost as much as before the pandemic, for example. This, in turn, buffered the economic impact on students and their families as well as on students' emotional and mental health. Likewise, our university is located in an area with an above average median household income, 90% percent or more of household have a computer, and 80% or more broadband internet connection (*United States Census Bureau, 2021*).

Following how students' behaviors and their emotional health change during academic year 2021/2022 will be very interesting. At the moment, the plan for Florida and our institution is to go back to pre-pandemic conditions as far as teaching, restrictions on class sizes, meetings, and indoors and outdoors social events (all removed), face masks

(not required anymore), and social distancing (not required anymore) are concerned. Unfortunately, because the State of Florida made it illegal to ask students, faculty, or staff to report their vaccination status (although the information can be provided voluntarily) or to require them to get vaccinated, what will or may happen is less certain than one may think. Among our survey respondents, only 64.2% (194 of 302 responses) indicated that the chances of them getting a COVID vaccination was 70% or higher, while more than one-quarter put the chances of them getting vaccinated at 30% or less (26.8%; 81 of 302 responses). These data indicate that our student population may still be vulnerable to COVID for some time, especially now that more contagious virus variants become dominant. By Spring 2022, seniors participating in our survey will have been freshmen or sophomores pre-pandemic and juniors during the pandemic, allowing us to chart their journey longitudinally.

## CONCLUSIONS

The longitudinal data collected and analyzed indicate that the impact of COVID-related restrictions on students' behaviors and habits as well as their mental and emotional health was less severe than one would have expected based on studies during the early stage of the pandemic. While some behaviors and habits changed during the COVID pandemic compared with the pre-COVID period, the changes were not substantial overall and most students' lives outside of studying went on with little change. Likewise, our study did not find an increase in the proportion of respondents considering, planning or attempting suicide during COVID, although study participants reported a significant increase in difficulty concentrating, remembering, or making decisions, as well as being impacted by feelings of sadness or hopelessness. Also, the cross-sectional data from our survey make the negative effect of the overall situation and the restrictions on students' mental and emotional well-being evident.

The impact of this pandemic will unquestionably be long-lasting and, thus, necessitate future investigations. Even though some countries and states in the US, including Florida, have dropped almost all restrictions and more or less plan on business as usual for schools, colleges, and universities for Fall 2021 and beyond, new waves of high infection rates caused by different SARS-CoV-2 variants may force a change of course. Studying how students react to even more or new restrictions and whether those who were impacted by the first round of restrictions will handle the situation better than those who enter higher education now, will be an interesting focus for further research.

### Funding

The authors received no funding for this work.

### Competing Interests

The authors declare there are no competing interests.

## Author Contributions

- Peter R. Reuter conceived and designed the experiments, performed the experiments, analyzed the data, prepared figures and/or tables, authored or reviewed drafts of the paper, and approved the final draft.
- Bridget L. Forster and Bethany J. Kruger conceived and designed the experiments, performed the experiments, authored or reviewed drafts of the paper, and approved the final draft.

## Ethics

The following information was supplied relating to ethical approvals (i.e., approving body and any reference numbers):

The Florida Gulf Coast University (FGCU) Institutional Review Board (IRB) approved the study protocol (March 30, 2018) and an amendment (October 29, 2020) prior to data collection.

## Data Availability

The raw survey (longitudinal and cross-sectional) data are available in the Supplementary File.

## Supplemental Information

Supplemental information for this article can be found online at http://dx.doi.org/10.7717/peerj.12528#supplemental-information.

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
