# Peer review of "A longitudinal study of the impact of COVID-19 restrictions on students' health behavior, mental health and emotional well-being"

_PeerJ, doi:10.7717/peerj.12528_

## Round 0.1 · original submission · Major Revisions

Your manuscript requires major revisions, particularly related to experimental design and how that impacts the conclusions. The reviewer comments appear to be clear and relevant, so please address their comments and modify your manuscript accordingly.

Reviewer 1 ·

Basic reporting

The article is clear and unambiguous. The background provided allows the reader to understand the overall situation during COVID in Florida and at Florida Gulf Coast University. It is well referenced throughout to support what is written. The structure of the article and the tables/data are clear. The results are detailed and address the hypothesis.

Experimental design

The research question is well defined, relevant and timely. The study provided needed information about the behaviors, health and wellbeing of college students during COVID-19 using a comparative study with pre-COVID 19 data. The authors intent to continue to chart students' journeys longitudinally will provide further essential detail on this topic. Publishing this article will allow others to replicate the study in other contexts. This is essential as each state in the US and countries globally have been impacted in different ways.

Validity of the findings

All underlying data has been provided and the finding/conclusions are well stated and link to the original research question around the behaviors, health and wellbeing of college students pre-and during COVID-19. They are clearly outlined and easily accessible for the reader to understand.

Additional comments

I think this is an excellent, relevant and timely article. The topic of the study is essential. It will be interesting to see the development in the study authors continue to further investigate the topic. On a personal note, I think it would be interesting to see a comparison of what has been done here to a state that had stricter lockdown measures/restrictions, such as California.

Reviewer 2 ·

Basic reporting

This manuscript provides unique information on how health outcomes and behaviors of young adults have changed from pre-pandemic to during the pandemic. In particular, the study data incorporates experiences of young adults that were collected over a one-year period of the pandemic whereas most prior studies have focused only on the initial few weeks or months of navigating precautions to slow COVID-19 transmission in the U.S. The methodology of the study reported in this manuscript has a number of limitations with regard to generalizability, but nonetheless publication of the results will make an important contribution to the literature. The authors should also be commended for the thorough review of literature related to their study results and the manner in which it is summarized to put their results into context. I have some suggestions for strengthening the manuscript with regard to presentation of the results and the clarity of some points made in the introduction and discussion. The authors do a strong job of discussing limitations of their data.
Lines 59-64. The text is a run-on sentence and needs editing to improve clarity.
Line 141. Need to spell out the acronym the first time it is used.
Lines 429-431. The reference comparison makes the sentence a little confusing.
Table 1. Please add titles for each column.
Table 2. Please add data to show effect size.

Experimental design

Lines 388-390. The overall low response rate should also be mentioned here.

Validity of the findings

Lines 210-213. The difference should be described more clearly in this sentence and the directionality of the difference should be indicated. It is not clear why the authors refer to a proportion. Similar issues with language occur throughout the results.
Line 262-264. This is another sentence that lacks directionality with regard to the reporting of differences. The entire results section should be reviewed to edit for clarity.
Lines 266-268. This line reads very well and is easy to understand – an example of how it would be nice to similarly reword other text in the results.
Lines 316-319. It would be helpful to start to the discussion with a brief restatement of the research questions and a summary of how the results are novel. This sentence does not describe the directionality of changes.

Reviewer 3 ·

Basic reporting

This reviewer has a few comments on Basic Reporting of the manuscript:

1) Currently within the Methods – subtitled COVID-19 related restrictions in Florida – Please clarify the years for each Phase of the state orders. Additionally, this information (related to the Florida orders) would be better fit in the Introduction section of the manuscript to offer context to the restrictions experienced by the target population.

Experimental design

This reviewer noted a number of comments and questions related to the methodological and design of the present study. Most importantly, the Methods section should be reorganized to follow standards similar to other observational studies of mental health. This reviewer suggests including relevant subheadings such as “Study Design”, “Participants”, “Measurement” to improve the clarity of the manuscript. For example, the Measurement subheading should include detailed information about the outcomes selected and reported in the Results section without having to jump to a table/Appendix.

1. Methods line 129 – would clarify that it is a bi-annual survey.

2. Response rate of 6-10% (line 153): This is a fairly low response rate – how do responders are different from non-responders?

3. Methods line 141 – Please add information what the acronym FGCU means.
Note: it is clarified later (line 160), but it should be added to the first time the acronym is introduced.

4. Lines 162 – 165: This demographic breakdown is interesting, but does not need to be discussed in this section. This reviewer would be more interested in findings related to how the analytic sample compares to the overall breakdown of the school (particularly with respect to responders and non-responders).

5. This reviewer suggests adding more information about how the survey was modified, and to what extent.

6. Were the samples the same cohort who were measured twice? Or were they two unique samples? It would be important to clarify this for the reader. Additionally, it would be important to note, if these were two unique samples, how they differed.
6a. If these are two unique groups, and the respondents were not matched across surveys, this should be revised to indicate that this is a repeated cross-sectional study that was measured at different timepoints and compared.
6b. If these surveys were matched to participants, the analysis needs to take into account the violation of independence and will need to be revised to account for this within-subject correlation.

7. How are the two groups defined? Are all surveys conducted prior to the COVID-19 pandemic included in Group 1?

Validity of the findings

The authors of this manuscript should temper the language of the implications and conclusions of these findings given that the study is not longitudinal, but rather (as presented) is a repeated cross-sectional study (i.e., the same survey measured with different samples). This is a major limitation of the current analysis and should be addressed thoroughly. This authors highlight the longitudinal nature of the study as a strength, but the analysis does not match this proposed design.

Results:

1. Line 204: Since the authors are comparing to the general student population, it would be useful to know the percentage of males and females in the sample (in addition to the reported statistical significance).

2. What is the definition of physical activity in the context of this study? Could be a result of less access, e.g., Pre-COVID cohort had access to gym but also could report increased physical activity because of in-person learning.

3. Please report results (percentages, means/SD, p-values) in-text, rather than referring to the Tables.

Discussion

1. Lines 334 – 338: Do the authors have additional data about this rationale? If so, this reviewer suggests adding this information to the Results. If this data is not available, this reviewer suggests tempering the proposed reasoning as it is outside of the scope of the current study.

2. Line 339 - 340: This reviewer is uncertain how the authors drew this conclusion based on the information presented.

3. Line 362: Given the cross-sectional nature of the data, the authors are unable to make a conclusion about the impact on food consumption and should revise the language of the manuscript to align with the scope of this work.

---

## Round 0.2 · Minor Revisions

One of the reviewers believes that you were not responsive to requests to clarify language regarding the study design. The discussion text needs editing to improve the interpretation of results based on a repeated cross-sectional survey design vs longitudinal design.

Reviewer 2 ·

Basic reporting

The text of the results section remains somewhat confusing. The authors were modestly responsive to requests with regards to improving ambiguous language, but the directionality of findings is not consistently indicated and the language regarding percentages is awkward in multiple sentences. It is very difficult to interpret sentences such as "The proportion of respondents reporting average hours of sleep over the past seven days differed between the pre-COVID group and the COVID group (Table 3)." The results text describing vaping, alcohol, and marijuana use is an example of strongly written and unambiguous language but the authors did not completely eliminate problematic language within other sections of the results. The authors provide some estimate of the magnitude of changes in behaviors with regard to substance use (i.e., getting at effect size) and for some other health outcomes but this is not done consistently throughout the results.

Experimental design

The details of COVID-related orders in Florida and at the FGCU is somewhat excessive and the key aspects of these orders is missing as context in framing the research questions within the introduction.

The authors were not responsive to requests to clarify language regarding the study design. The discussion text needs extensive editing to improve the interpretation of results based on a repeated cross-sectional survey design vs longitudinal design.

Validity of the findings

The conclusions are difficult to understand because they reference the severity of outcomes reported by other studies and no direct comparisons can be validly made to the effect sizes reported within the current study.

---

## Round 0.3 · accepted · Accept

Thank you for addressing the final round of reviewer comments.